



# Evaluating ocean alkalinity enhancement for carbon dioxide removal: evidence from a one-year saltmarsh field experiment

Isabel Mendes[1], Julia Lübbers[1], Joachim Schönfeld[2], Alexandra Cravo[1]

[1] Centro de Investigação Marinha e Ambiental (CIMA), Rede de infraestrutura em Recursos Aquáticos (ARNET), Universidade do Algarve, Campus de Gambelas, Edifício 7, 8005-139 Faro, Portugal
[2] GEOMAR Helmholtz Centre for Ocean Research Kiel, Wischhofstrasse 1-3, 24148 Kiel, Germany

*Correspondence to*: Isabel Mendes (imendes@ualg.pt)

**Abstract.** Ocean alkalinity enhancement is a promising carbon dioxide removal (CDR) strategy aimed at reducing atmospheric carbon dioxide ($CO_2$) concentrations. To evaluate its effectiveness and potential biogeochemical impacts, field experiments under natural conditions are essential. We report results from a one-year in-situ experiment conducted in the saltmarsh pioneer vegetation zone at Ria Formosa coastal lagoon, Portugal. The experiment comprised replicate deployments of olivine and basalt (treatments), and untreated control sites. Total alkalinity (TA) responded immediately to the treatments, with pore water 1.5 to 2.3 mM higher than the control. High concentrations of $CO_2$ in pore water led to an increase of dissolved inorganic carbon (DIC) higher than TA. Continuous $CO_2$ degassing from the saltmarsh soil was observed, with the treatments prompting higher $CO_2$ fluxes than control. Carbon was laterally exported to the ocean (outwelling), following the trend of excess TA production. This effect was most pronounced during the first seven months after deployment, with basalt producing the best results. These findings provide critical insights into the temporal dynamics and efficacy of alkalinity enhancement in coastal vegetated systems.

## 1 Introduction

Reducing atmospheric carbon dioxide ($CO_2$) concentrations to combat global warming and associated environmental changes is one of the greatest challenges of humanity (IPCC, 2018). To prevent a pervasive rise temperature of more than 1.5°C above pre-industrial levels, technical measures are necessary to remove more $CO_2$ from the atmosphere (Carbon Dioxide Removal, CDR), and to assure its long-term storage (IPCC, 2023). Ocean alkalinity enhancement (OAE) is one of the most promising CDR measures, with a high potential to increase oceanic carbon uptake and storage (Kheshgi, 1995; Foteinis et al. 2022; Yang et al. 2023). An increase of seawater alkalinity will bind more atmospheric $CO_2$ as bicarbonate and carbonate ions, and consequently increase the pH (Kheshgi, 1995; Feng et al., 2017; Moras et al. 2022). This process can be fuelled by spreading mafic rocks and minerals (e.g., basalt, olivine), lime, limestone powder, or synthetic minerals in marine environments (e.g., Köhler et al., 2013; Montserrat et al., 2017; Bach et al., 2019; Moras et al., 2022). The reactions of these substrates with



seawater have been tested with laboratory experiments or by modelling approaches (e.g., Monserrat et al., 2017; Bach et al., 2019; Fuhr et al., 2022). The data revealed that the atmospheric $CO_2$ consumption during mineral dissolution depends on the minerals used, their grain size, temperature, pH, and other abiotic parameters (Hangx and Spiers, 2009; Hartmann et al., 2013; Feng et al., 2017; Geerts et al., 2025). The results demonstrated a promising potential for $CO_2$ uptake, but the ensuing

biogeochemical changes, i.e. trace metal or nutrient release of mafic minerals (Bach et al., 2019), and the ecological impacts of alkalinity enhancement on the environment are yet unconstrained (e.g., Meysman and Monserrat, 2017; Cyronak et al., 2023). Only a few field experiments have been conducted under natural conditions in open waters (e.g., Albright et al., 2016; Voosen et al., 2022), facing technical constraints, detection thresholds, or ambiguous effects (Tollefson, 2016). Experiments in near coastal settings under stable environmental conditions, which are readily accessible and regularly monitored to assess

the underlying physical and ecological variability, including untreated control sites, were considered to provide more precise estimates of atmospheric $CO_2$ uptake and realistic scenarios of the ecological and biogeochemical dynamics (Cyronak et al., 2023; He and Tyka, 2023). However, very few studies are available. Tidal wetlands, mangroves and salt marshes already act as natural sinks for atmospheric $CO_2$, sequestering carbon in their biomass and sediments. During ebbing tidal phase, some of this carbon can also be exported laterally through pore water exchange (outwelling), which involves the transport of dissolved

inorganic carbon (DIC is the sum of dissolved $CO_2$, bicarbonate ($HCO_3^-$), and carbonate ($CO_3^{2-}$) ions) and total alkalinity (TA) to the ocean, contributing to long-term carbon storage (e.g., Chen et al., 2021; Reithmaier et al., 2023, Yau et al., 2022).

The Ria Formosa coastal lagoon, offers the ideal conditions to perform an OAE experiment, running under natural conditions. Located on the southern coast of Portugal, this shallow mesotidal lagoon is characterized by a Mediterranean climate. The lagoon has a subtidal and intertidal surface area of 88 km2, composed of tidal channels, tidal flats and saltmarsh with different

types of vegetation, according to the level and duration of tidal submersion (e.g., Schönfeld and Mendes, 2021; Carrasco et al., 2021) (Fig. 1). The mean water depth was estimated to 3 m (e.g., Dias and Sousa, 2009; Carrasco et al., 2018). This open natural system has five inlets between barrier islands, with a mean, semi-diurnal tidal range of 1.3 m to 3.5 m, facilitating twice a day the exchange of water and nutrients with the Gulf of Cadiz (Atlantic Ocean) (e.g., Pacheco et al., 2010; Jacob and Cravo, 2019; Cravo et al., 2024). The pioneer vegetation zone is submerged during 23 to 50 % of the observation time of one lunar

year (Schönfeld and Mendes, 2022). The salinity and temperature of intertidal waters ranged from 15 to 39 and 9 to 30°C, respectively (Barbosa, 2010; Schönfeld and Mendes, 2021). Freshwater input to the lagoon is limited and primarily driven by winter precipitation (Barbosa, 2010). Sediments in the intertidal area consist mainly of mud, with a silt content generally above 50 %, and clay between 10 % and 20 % in tidal flats and saltmarshes, while coarse sand is lower than 5 % and occur in deeper parts of main channels (Andrade et al., 2004). Data from previous studies showed that high total alkalinity values were recorded

inside the lagoon (~2.21 mM) as compared to the adjacent Atlantic (~2.17 mM) (Schönfeld and Mendes, 2022). The gradient depicted a persistent export of alkalinity during ebb cycles. The average volume of the ebb phase was estimated to 7.305 x 107 $m^3$ (Pacheco et al., 2010), which carried an alkalinity export of 2.186 Gmol yr$^{-1}$, representing to a $CO_2$ export potentially up to 8.08 x 10$^4$ ton yr$^{-1}$ by the Ria Formosa tidal system under present, natural conditions.




Here, we compile a one-year dataset from the first European field experiment conducted in the pioneer vegetation saltmarsh
of the Ria Formosa coastal lagoon, Portugal, to evaluate the effectiveness of ocean alkalinity enhancement (OAE) using olivine
and basalt. Total alkalinity (TA), pH, temperature and salinity were measured in pore water and surface water (i.e. supernatant
water in puddles after emergence during lowering tide) from 4 treatments (with olivine (O) and basalt (B), each with fine (F)
and coarse (C) grain sizes) and 1 control site, with 3 replicates per condition (Fig. 1c). Dissolved inorganic carbon (DIC), ratio
of TA:DIC, fluxes between the air, surface water, and pore water, and TA and DIC lateral export were estimated to evaluate
the efficacy of OAE under these experimental settings.

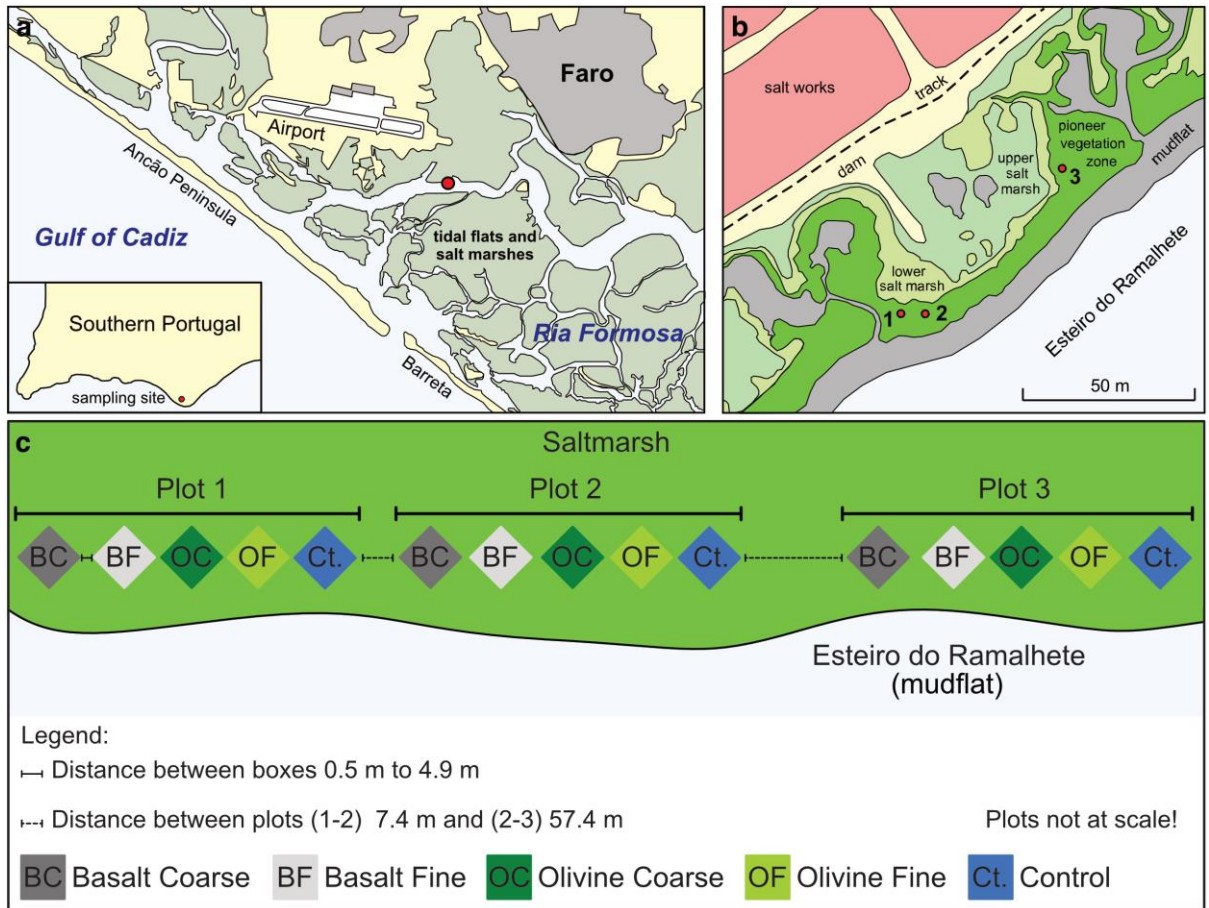

**Figure 1: Location of the Ria Formosa and the experimental site (red dot) (a), the location of the three plots in the pioneer vegetation zone (3 red dots) (b) and the schematic experimental setup with the plots, each one with the five experimental boxes (c). Adapted from Carta Militar de Portugal, Folha 610, 611, Instituto Geográfico do Exército, Lisboa, 2006 (a, b).**



## 2 Material and methods

### 2.1 Experimental setup and installation

The experiment was conducted in situ under natural conditions in a coastal lagoon salt marsh, characterized by a low-energy environment. Three replicate plots at distances of 7.4 and 57.4 m were place within a homogenous *Spartina maritima* thicket in the pioneer vegetation zone of Ria Formosa salt marsh (37° 0.36'N, 7° 57.89'W) (Fig. 1). Each replicate plot with five experimental boxes were placed between 0.29–0.85 m above Portuguese Ordnance Datum, which is submerged between 20 to 38 % of the time during the tidal cycle (24 hours). The boxes were spaced 0.5–4.9 m apart and framed by untreated pinewood (60 cm × 60 cm), positioned with 7–8 cm embedded in the sediment and 2–3 cm extending above it. Frames were placed horizontally, with the diagonal axis perpendicular to the marsh slope. One box per plot was left barren as a control. Adjacent boxes were covered with: a 0.5-cm layer of Norwegian olivine (picro-basalt), grades "AFS 120" (90–180 µm, "Olivine Fine") and "AFS 50" (180–355 µm, "Olivine Coarse"). A 1-cm layer of German basalt (basanite), fractions "Durubas" (63–355 µm, "Basalt Fine") and "Basalt Fugensand" (0.5–1.0 mm, "Basalt Coarse") (for further details see Mendes et al., 2025).

### 2.2 Water sampling and on-site measurements

Pore water from inside the substrates, surface water (supernatant) in contact with the substrates both in the experimental boxes, and tidal water from Esteiro do Ramalhete, were sampled monthly, starting the day after substrate deployment. Sampling occurred during the ebb period of the first low tide after sunrise, once sites emerged. Water samples were collected using pre-cleaned, acid-decontaminated (10% HCl) vials or syringes and stored in 20 - ml Zinsser vials or 100 - ml polypropylene beakers. Pore water samples were extracted with rhizons (pore size: 0.1 µm) (Seeberg-Elverfeldt et al., 2005; Schönfeld and Mendes, 2022), inserted horizontally into the 0.5–1 cm surface layer. Four to five rhizons per box collected ~15 ml of water per hour per rhizon, which was pooled and distributed for analysis. Samples for alkalinity, nutrients, and metals were immediately transported on ice to the University of Algarve laboratory. Sediment temperature was measured in each box with a plug-in thermometer. Salinity, oxygen, and pH of pore, surface, and tidal water were measured on-site using a WTW Multi 3620 IDS multiparameter probe. The pH sensor was calibrated with buffer solutions (pH 4: YSI-381), pH 7: YSI-3822, and pH 10: YSI-3823) on the total pH scale. According to the manufacturer's, the buffer solutions have an accuracy of ±0.002 units at 25°C. The conductivity sensor (TetraCon® 925) was calibrated with a KCl solution (1413 µS cm⁻¹ at 25°C; accuracy ±0.1) (for further details see Mendes et al., 2025).

### 2.3 Laboratory analyses

The total alkalinity of the water samples was measured on the sampling day by using Bruevich's titration method (Wallmann et al., 2006; Pavlova et al., 2008). An IAPSO standard with a total alkalinity of 2.325 mmol l⁻¹ was measured in triplicate before and after a sample series. Each water sample was analysed in triplicate too. The measured values from the samples were corrected with the target value of the IAPSO standard, and the total alkalinity was calculated with standard formulas (Wallmann



et al., 2023). The accuracy of alkalinity measurements was constrained by measurements of the Dickson Alkalinity Standard to ±0.0036 µM (1-sigma) (for further details see Mendes et al., 2025). The carbonate system parameters, e.g., pCO₂, $CO_2$, $HCO_3^-$, $CO_3^{2-}$ and DIC were calculated from the total alkalinity, pH, temperature, salinity, and nutrient concentrations by using the CO2SYS software, version 25b06 (Lewis and Wallace, 1998).

**110    2.4 Water - atmosphere CO₂ flux calculations**

The flux of $CO_2$ between surface water and the atmosphere (FCO₂, in mmol m$^{-2}$ d$^{-1}$) was calculated using the gas exchange formula (Wanninkhof et al., 2009):

$$FCO_2 = k \, K_0 \, (pCO_{2Sea} - pCO_{2Air}) \tag{1}$$

where $k$ (cm h$^{-1}$) is the gas transfer velocity, $K_0$ (mol L$^{-1}$ atm$^{-1}$) is $CO_2$ solubility in seawater, $pCO_{2Sea}$ and $pCO_{2Air}$ are the

respectively partial pressure of $CO_2$ in the ocean and atmosphere. $pCO_{2Sea}$ was derived using the CO2SYS program based on our sample data, while $pCO_{2Air}$ represents the globally averaged monthly mean $CO_2$ concentration over marine surface sites, provided by NOAA's monitoring program (https://gml.noaa.gov/ccgg/trends/global.html). A positive FCO₂ value indicating a flux from the water to the atmosphere, while a negative FCO₂ value indicates the ocean is acting as a $CO_2$ sink to the atmosphere.

The gas transfer velocity ($k$) was calculated using the wind speed parameterization proposed for coastal systems at 10 m high ($U_{10}$) (Jiang et al., 2008):

$$k = (0.314 \, U_{10}^2 - 0.436 U_{10}^2 + 3.990) * Sc/660^{-0.5} \tag{2}$$

Where $U_{10}$ is the average windspeed (in ms$^{-1}$) over the 4-hour sampling interval. The wind data, measured in 10 minutes intervals at the adjacent Faro airport was provided by the Instituto Português do Mar e da Atmosfera. Sc is the Schmidt number

and is specific to $CO_2$ which is calculated, for salinity between 0 and 35, as a function of the temperature using the expression (Jiang et al., 2008):

Schmidt number for salt water:

$$Sc_{fw} = 2116.8 - 136.25(SST) + 4.7353(SST)^2 - 0.092307(SST)^3 + 0.0007555(SST)^4 \tag{3}$$

Schmidt number for fresh water:

$$Sc_{sw} = 1923.6 - 125.06(SST) + 4.3773(SST)^2 - 0.092307(SST)^3 + 0.0007555(SST)^4 \tag{4}$$

To compensate for the higher salinity in our study and under the assumption that SC varies in linear way with the salinity, we calculated the Sc for the higher salinity measured in our study area,

$$Sc = Sc^{fw} + ((Sc_{sw} - Sc_{fw})/35) * SSS \tag{5}$$

In situ sea surface temperature (SST) and sea surface salinity (SSS) was directly derived from our samples.

$CO_2$ solubility ($K_0$) is a function of the SST and SSS (Weiss, 1974):

$$Ln(K_0) = A1 + A2(100/SST) + A3*In(SST/100) + SSS[B1 + B2(SST/100) + B3(SST/100)^2] \tag{6}$$

where A1 = −58.0931, A2 = 90.5069, A3 = 22.2940, B1 = 0.027766, B2 = −0.025888, and B3 = 0.0050578 are $CO_2$-specific coefficients.





### 2.5 Supernatant - pore water $CO_2$ flux calculations

The diffuse fluxes of supernatant - pore water gases from sediments were calculated using Fick's first law of diffusion:

$J = - \varphi Ds \ (dc/dz)$, (mg m$^{-2}$ d$^{-1}$)                                                                                          (7)

Where J is the diffusive flux, $\varphi$ porosity, Ds the sediment diffusion coefficient for each individual gas, and dc/dz the concentration change for each gas with depth. The concentration gradient was determined by comparing the $CO_2$ concentration in the surface water, just above the sediment-water interface, with the $CO_2$ concentrations in the pore water within the top 1

cm of the sediment. The calculations were therefore performed using a vertical distance of 1 cm. A positive diffusion flux (J) indicating a flux from the pore water to the surface water, while a negative flux indicates the pore water is acting as a $CO_2$ sink for the supernatant water.

The porosity ($\varphi$) was determined for the top 2 cm of sediment in each box by measuring the weight loss after drying the sediment at 75°C for three days. The samples had a known weight and volume. Quarterly porosity values were calculated and

averaged over the first year for subsequent analyses.

The sediment diffusion coefficient was then determined for fine-grained marine sediment from the porosity (Rasmussen and Jørgensen, 1992):

$Ds=D_0/(1+3(1-\varphi))$                                                                                          (8)

Following Rasmussen and Jørgensen (1992), the original relationship proposed by Ullman and Aller (1982), $Ds=D_0 \ \varphi^2$, was

found to produce Ds values that were approximately 5% higher than those generated by their proposed equation.

$D_0$ (m$^2$/s) is the molecular diffusion coefficient can be calculated using the Stokes-Einstein Equation:

$D_0=(\kappa_B*T)/(6 \ \pi \ \eta \ r)$                                                                                          (9)

where $\kappa_B$ is Boltzmann's constant (1.38*10$^{-23}$ JK$^{-1}$), T is the absolute temperature (K), $\eta$ is the dynamic viscosity (Pa*S), and r is the radius of the $CO_2$ molecule (m).

As flux occurred only during submersion of the salt marsh, calculations were based on an average submersion time of 30%. Submersion time at the plot ranged from 20% to 38%, with a mean of 30%.

### 2.6 DIC and TA outwelling calculation

The DIC and TA outwelling rates were calculated monthly from September 2022 to September 2023 for the first ebb tide of the sampling day when there was hydrological export of water to the open ocean. The export/import was calculated after

Neubauer and Anderson (2003):

$DIC/TA_{export}=([(((DIC/TA_{sw})/h) + ((DIC/TA_{tide})/h))/2] * (V_{sw}-V_{tide}))/A$, (mmol m$^{-2}$ h$^{-1}$)                                       (10)

The surface water volume ($V_{sw}$) was determined from the box dimensions (60 cm × 60 cm) and a water depth of 2.5 cm, corresponding to the conditions immediately after emergence, when 2–3 cm of water remained trapped inside the boxes. The rising tide volume ($V_{tide}$) was similarly determined from the dimensions of the box and a water depth of 5 cm, corresponding





to the depth at which the boxes are completely submerged, as tide water samples were taken outside the boxes, close to the plots, while the boxes were submerged. For the area (A), the area of the individual boxes (0.36 m²) was used.

The interval (h) represents the time for the rising tide to increase from box height to complete submersion with 2 cm of overlying water (from 2,5 to 5 cm water depth), set to 30 minutes based on over a year of site observations (>13 visits).

## 2.7 Carbon dioxide mass calculations

The annual carbon dioxide ($CO_2$) mass budget was derived from the flux calculations for each treatment and each flux (surface water–air and surface water–pore water). The fluxes were multiplied by the area of the boxes, considering the sum of the three boxes of 60 cm × 60 cm each. The results were then multiplied by the molecular weight of $CO_2$ (44 mg mmol$^{-1}$). This provides the $CO_2$ mass in mg per day for each treatment and sampling event.

For each treatment, the results from all sampling events were summed and then divided by the number of sampling times to 180 obtain an average mass per day. This average was subsequently multiplied by 365 to estimate the annual mass.

The treatments were then compared with the control (treatment minus control) to determine which treatment was more efficient. This calculation was carried out separately for pore water and supernatant water.

## 2.8 Statistical analysis

A two-way analysis of variance (ANOVA) was conducted to assess the effects of the two treatments in comparison to natural 185 conditions and to evaluate variability among the three replicates (plots). Additionally, a two-way ANOVA was used to examine the interaction between treatment type and grain size (fine and coarse). To directly compare the two treatments, two-sample t-tests were performed. All statistical analyses were carried out using PAST software (PAlaeontological STatistics, ver. 4 (Hammer et al., 2001)) and Microsoft Excel.

## 3 Results

### 3.1 Environmental variability

The daily air temperatures recorded at Faro airport close to the experimental area (~1.5 km to the northwest), varied from 3.2 to 25.7 °C (daily minimum) and from 13 to 38.7 °C (daily maximum), between September 2022 and September 2023. The temperatures followed the typical seasonal cycle, with low temperatures from January to March, and high temperatures from June to August 2023 (Fig. 2a). Mean channel water temperatures showed the same pattern, ranging from 12.5 to 22.7 °C during 195 the lowering (ebb) tide and from 15.8 to 27.1 °C during the rising (flood) tide, with higher values resulting from solar heating of the exposed mudflat during the preceding low tide. The saltmarsh soil temperature recorded at the control boxes resembles the maximum air temperature, ranging from 17.7 to 31.2 °C (Fig. 2a). For the same period, mean precipitation was low (0.9 mm/day), mostly occurring from October to March, with only 10 consecutive days of precipitation in December 2022 yielding 158 mm (Fig. 2b). The lowest salinity was also recorded in December in all the water samples replicates (33–34) reflecting



the recorded volume of precipitation. The highest salinity was recorded in pore water samples during the warm period, in August, probably reflecting higher evaporation rates (39.5). In general, except in the periods of rainfall, the salinity values of water samples from surface control boxes and channel during lowering tide were similar (36–37.6) (Fig. 2b), and the same was observed for the pH values (7.7–8.3) (Fig. 2c). The lowest pH was recorded in pore water from the control sites (6.7–7.3) in December, when the rainfall was at maximum as well as in late spring – summer. In contrast, the highest pH (8.4–8.9)

occurred in channel water during rising tide in February and March, likely driven by observed diatom mats and enhanced photosynthetic activity. The highest variation of mean total alkalinity (TA) was observed in pore water from control boxes, with the maximum in September 2022 (2.9 mM) and the minimum in January 2023 (1.8 mM). The lowest variation was observed in control surface water with TA ranging from 2.3 mM in May to 2.9 mM in July 2023. A general trend of higher TA in the surface water than in the rising tide, was also observed (Fig. 2d). No temporal and spatial trend was evident.









**Figure 2: Environmental parameters variability at the experimental site, from September 2022 to September 2023. Data were collected from water samples in the Esteiro do Ramalhete channel during both lowering (ebb) and rising tides (flood), as well as in the surface water and pore water in the control boxes (means of three replicates). a) Temperature variation in the channel water and in the saltmarsh soil (in situ). Minimum and maximum air temperatures are shown for comparison; b) Salinity variation in**
**water samples and daily precipitation; c) pH and d) total alkalinity in control box water samples, for comparison with the treatments. Air temperature and precipitation data from Faro airport (~1.5 km from the study area) provided by Instituto Português do Mar e da Atmosfera, Lisbon, Portugal (https://www.ipma.pt/pt/oclima/monitoriza.dia/).**

### 3.2 Alkalinity enhancement

Considering the surface water samples taken along one year (Sep. 2022 – Sep. 2023), the highest variations of total alkalinity
(2.08 - 3.22 mM) were recorded in September 2022 on the day after deployment of olivine and basalt substrates. During the rest of the year, TA variation from 0.7 mM (Dec. 2022) to 0.2 mM (Aug. and Sep. 2023) were observed, with values in the treatments similar ($p > 0.05$) to those of the control boxes (Fig. 3a). Excess alkalinity (difference between the mean alkalinity from each treatment and mean alkalinity from the control over all 3 plots) in the surface water on the experimental substrates showed generally high values during the first seven months of the experiment for all the treatments (Sep. 2022 – Mar. 2023).
The highest average values of excess alkalinity were recorded in Sep. 2022, with 0.5 mM for Basalt Coarse, 0.3 mM for Basalt Fine, 0.2 mM for Olivine Coarse, and no variation for Olivine Fine (Supplementary Fig. S1)

In the pore water, TA followed a similar behaviour as in the surface water with a more accentuated alkalinity enhancement, increasing to values around 5 mM for all the treatments on the day after deployment (Fig. 3b). At this time, the excess alkalinity was higher by 1.5 to 2.3 mM and decreased, in an exponential manner in all treatments (Supplementary Fig. S2), to reach
typical ambient values in April 2023, around mean values of 2.5 mM (Fig. 3b). The alkalinity was in general higher in the fine treatments (olivine 2.1 and basalt 2.3 mM) as compared to the coarse treatments (olivine 1.5 and basalt 1.9 mM) (Fig. 3b). This mirrors a pervasive gradient, and hence alkalinity flux from the weathering substrate to the pore water, which is more intense after substrate deployment and effective until at least April 2023.



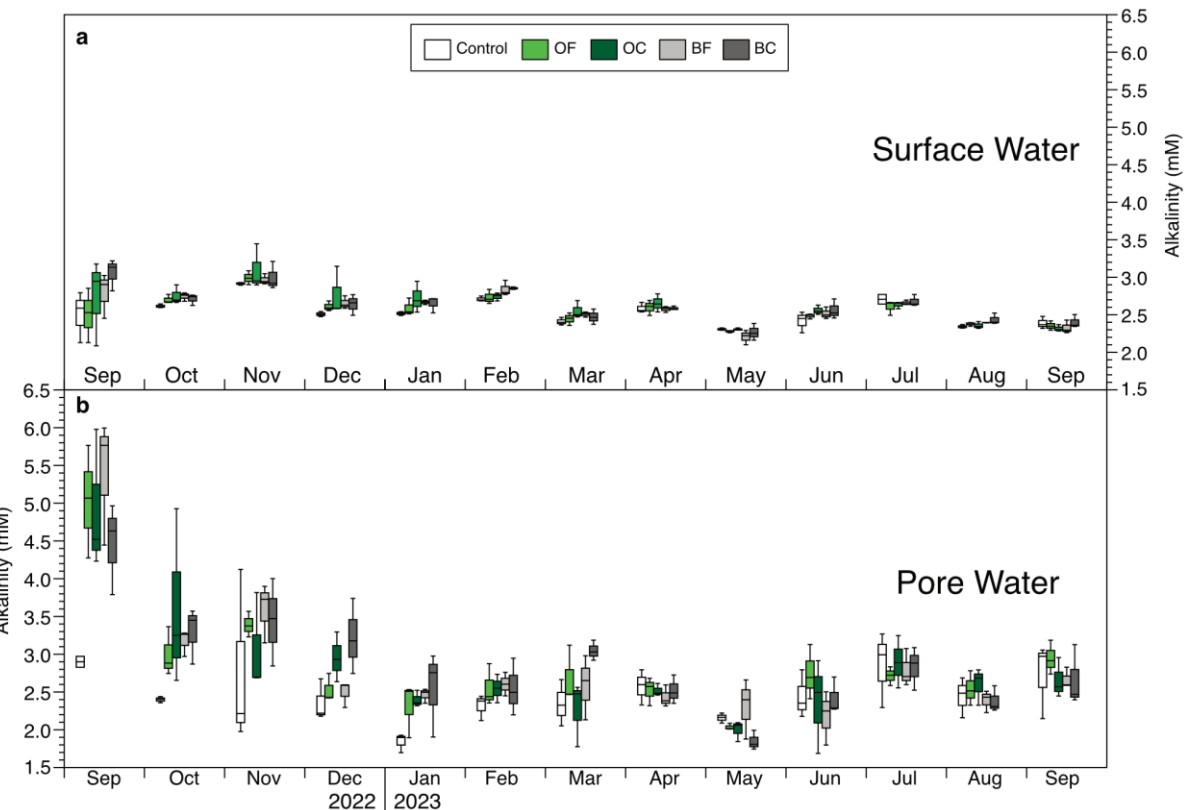

**Figure 3: Alkalinity variation in the experimental site from September 2022 to September 2023. a) Surface water and b) Pore water alkalinity (mM) boxplots monthly variation in the control and each applied treatment. Monthly boxplots represent individual treatments in the following order: Control (white), Olivine Fine (OF, light green), Olivine Coarse (OC, dark green), Basalt Fine (BF, light grey), and Basalt Coarse (BC, dark grey). The boxplots indicate the median (middle line), 25th and 75th percentile (box) and the minimum and maximum values (whiskers). (Excess alkalinity presented in Supplementary Figs. S1, S2).**

## 3.3 TA:DIC ratios

The TA:DIC ratio reflects the balance between the buffering capacity of seawater (TA) and the amount of carbon available for acid-base reactions (DIC) and allows an assessment of the vulnerability of an ecosystem to acidification. Low TA:DIC ratios indicate a reduced ability to buffer pH changes, which can occur when more atmospheric $CO_2$ is absorbed (Egleston et al., 2010; Reithmaier et al., 2023). Respiration is also important as it leads to the production of $CO_2$ that is accounted for DIC calculation but not for TA, therefore also lowering the ratio TA:DIC.

The TA:DIC ratios in the surface water generally showed values higher than 1 from September 2022 to September 2023. Similar ratios ($p > 0.05$) were observed among the control boxes (minimum 1.02 to maximum 1.19) and the different treatments (0.98 to 1.18). The higher values were recorded in February 2023 and August 2023 (average 1.15±0.02) and the lowest in December 2022 (1.03±0.02) (Fig. 4a). In the pore waters, TA:DIC ratio variability pattern shifted and showed mostly values lower than 1, indicating higher concentrations of DIC than TA, and a decreasing trend from January to May 2023. Persistent lower values were recorded from May to September 2023 (Fig. 4b). In the control boxes, values ranged from 0.73 to 1.07, with





0.89±0.07 on average. In the treatment boxes, the TA:DIC ratios ranged from 0.68 to 1.14 with mean values ranging from 0.89±0.06 for olivine coarse and 0.91±0.07 for both basalt treatments. Values higher than 1 were only observed in September 2022 (1.04 to 1.10), just after the deployment of the substrates for all the treatments, and in January 2023 for basalt fine (0.93

to 1.14). TA:DIC ratios of lowering tide (Fig. 4c) resemble the ratios of control surface water samples, after six hours of high tide cover of the saltmarsh. The rising tide of Atlantic water showed higher values of TA:DIC ratios compared to the outflowing water during lowering tide, with pronounced seasonal variability throughout the year.

**Figure 4: TA:DIC ratio variation at the experimental site from September 2022 to September 2023. TA:DIC ratio boxplots display**
**the monthly variation in the control and each applied treatment in a) surface water, b) pore water and c) tidal channel water during lowering (ebb) and rising (flood) tide (see Fig. 3 for boxplot information). DIC was calculated with the CO2SYS program.**




### 3.4 Fluxes of CO₂ atmosphere - water and pore water - surface water

The influence of olivine and basalt on carbon exchange processes at the experimental site, was evaluated through $CO_2$ fluxes across atmosphere–surface water and pore water–surface water (Fig. 5). The trend of these fluxes closely reflected the partial

pressure of $CO_2$ ($pCO_2$) for surface and porewater samples (Supplementary Fig. S3). Overall, $CO_2$ fluxes between the atmosphere and surface water were positive during the experimental period. The values were similar ($p > 0.05$) between the control and the treatments. Positive values were observed in the first four months of the experiment, and in June, July and September 2023, demonstrating that the surface water is a substantial source of $CO_2$ to the atmosphere. The highest fluxes were recorded in December 2022 in both, the control ($53.37\pm11.03$ mmol m$^{-2}$ d$^{-1}$) and treatments ($62.31\pm41.26$ mmol m$^{-2}$ d$^{-1}$),

with maximum degassing through the layer of olivine coarse (185.19 mmol m$^{-2}$ d$^{-1}$). February was the only month when $CO_2$ was taken up from the atmosphere in the control ($-17.4\pm4.2$ mmol m$^{-2}$ d$^{-1}$) and in the treatments ($-16.4\pm3.5$ mmol m$^{-2}$ d$^{-1}$) (Fig. 5a).

The $CO_2$ fluxes from the pore water to the surface water, except in September and December 2022 were higher than from the surface water to the atmosphere, indicating a continuous degassing from the salt marsh soil to the supernatant water throughout

the year. The lowest fluxes values in all the treatments (mean $1.36\pm0.76$ mmol m$^{-2}$ d$^{-1}$) compared to the control ($10.74\pm4.43$ mmol m$^{-2}$ d$^{-1}$) were recorded in September 2022, just after the substrate deployment. The higher ($p < 0.05$) fluxes in the treatments compared with the control were recorded in July 2023 for olivine fine and coarse, and for basalt coarse (average of the 3 treatments $28.27\pm6.37$ mmol m$^{-2}$ d$^{-1}$; control $14.56\pm0.23$ mmol m$^{-2}$ d$^{-1}$). Overall and throughout the year, olivine treatments effected generally higher $CO_2$ fluxes from the pore water to surface water than basalt (Fig. 5b).

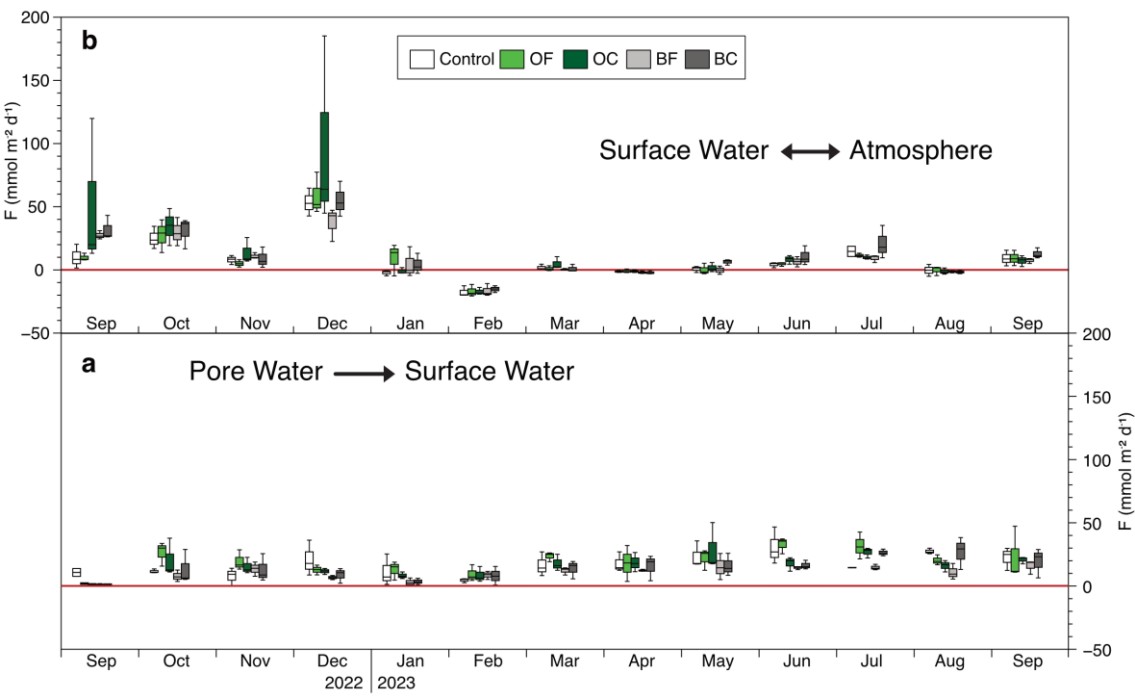



**Figure 5: Variation of carbon dioxide ($CO_2$) fluxes at the experimental site from September 2022 to September 2023. a) Fluxes of $CO_2$ between atmosphere and surface water, b) fluxes of $CO_2$ between pore water and surface water. Values above zero indicate degassing, while values below represent uptake. The box plots represent individual treatments (see legend of Fig. 3 for boxplots information).**

### 3.5 TA and DIC outwelling

Outwelling, i.e. the lateral export of carbon from intertidal areas to the ocean, was calculated based on the experimental box size (60 cm × 60 cm), for each of the treatments and controls and represents the mean of the 3 replicates surface water, considering the first ebb tide of the sampling day when there was hydrological export of water. TA and DIC showed a similar trend throughout the year (Fig. 6). Total alkalinity outwelling in the control boxes was on average $2971.1\pm176.5$ mmol m$^{-2}$ d$^{-1}$, with the lowest (2630.1 mmol m$^{-2}$ d$^{-1}$) and highest (3351.2 mmol m$^{-2}$ d$^{-1}$) values recorded in May 2023 and November 2022, respectively. From September 2022 to March 2023, TA outwelling slightly increased in the treatments (average of all treatments $3131.6\pm154.8$ mmol m$^{-2}$ d$^{-1}$) compared at the control sites ($3050.3\pm134.9$ mmol m$^{-2}$ d$^{-1}$). Seven months after the substrates' deployment, TA variation between the treatments and control decreased progressively, with average values between April to September 2023 ($2893.6\pm183.1$ mmol m$^{-2}$ d$^{-1}$) similar to the control ($2873.2\pm175.5$ mmol m$^{-2}$ d$^{-1}$) (Fig. 6a).

DIC outwelling, globally, showed the same trend as TA outwelling. In the control boxes was on average $2586.5\pm164.9$ mmol m$^{-2}$ d$^{-1}$ in, with mean minimum values attained during the same month, May 2023 (2218.1 mmol m$^{-2}$ d$^{-1}$) and the maximum value in September 2022 (2857.1 mmol m$^{-2}$ d$^{-1}$). From September 2022 to March 2023, DIC outwelling was on average higher in the treatments (average of all treatments $2679.8\pm226$ mmol m$^{-2}$ d$^{-1}$) than at the control sites ($2586.5\pm164.9$ mmol m$^{-2}$ d$^{-1}$). DIC outwelling also decreased progressively by time in the treatments (Apr. – Sep. 2023) to values close to the control. In this period, the average outwelling values of the control ($2405\pm157.8$ mmol m$^{-2}$ d$^{-1}$) and the treatments ($2427.1\pm175.6$ mmol m$^{-2}$ d$^{-1}$) were similar and showed the same temporal variation (Fig. 6b).



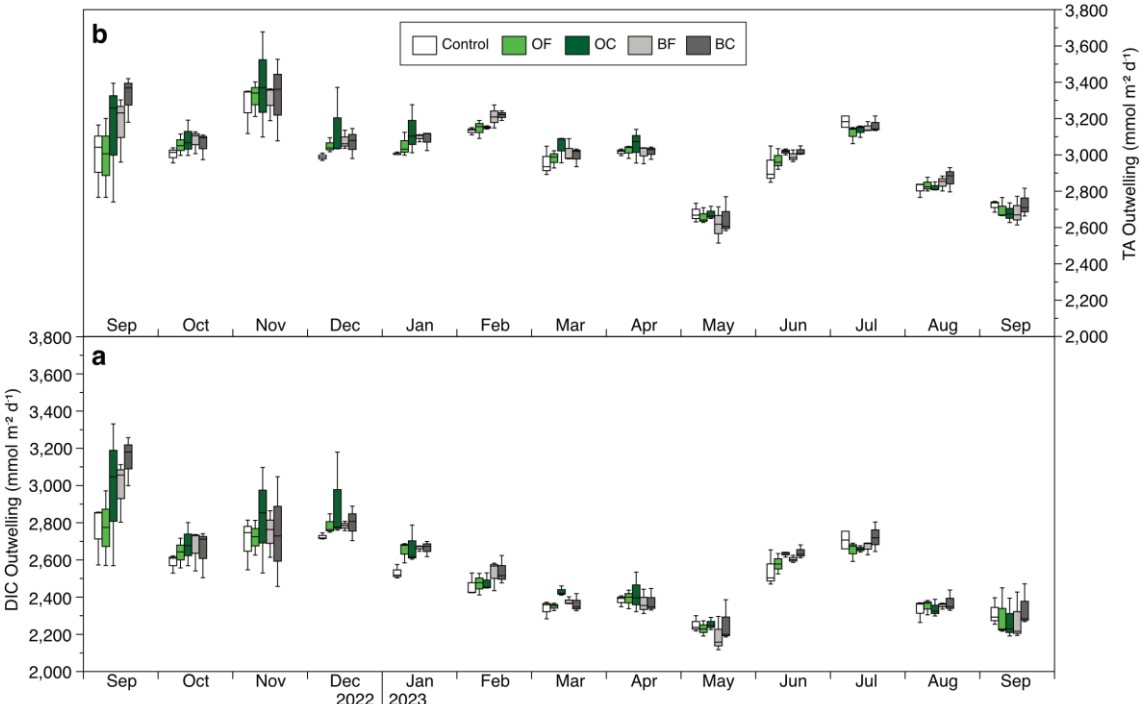

**Figure 6: Outwelling of Total Alkalinity (TA) (a) and dissolved Inorganic Carbon (DIC) (b) at the experimental site from September 2022 to September 2023. The box plots represent individual treatments (see legend of Fig. 3 for boxplot information).**

## 3.6 Estimation of the annual mass production of $CO_2$

The mass of carbon dioxide dissolved in the surface water-atmosphere exchange, based on the mean fluxes for the three replicates for the control and treatment boxes, extrapolated for one year of experimental duration, showed the lowest values in the control boxes (134.89 $CO_2$ g yr$^{-1}$) (Fig. 7, Supplementary Tab. S1). The difference between the treatments and the control was positive, indicating that all the surface water from the treatments produced more $CO_2$ than the control. The coarse-grained substrates of olivine (275.13 $CO_2$ g yr$^{-1}$) and basalt (210.79 $CO_2$ g yr$^{-1}$) were the treatments with higher $CO_2$ content (OC 51 % and BC 36 %), indicating they were outgassing to the atmosphere or outwelling more $CO_2$ than the control (Fig. 7a).

In the pore water, the mass of carbon dioxide in the control (282.23 $CO_2$ g yr$^{-1}$) was 2 times higher than in the surface water. The difference of treatments and control sites showed positive values for olivine fine and negative values for the other treatments, indicating that pore waters from olivine fine produced more $CO_2$ (13 %) than the control sites, while the other treatments, mainly basalt fine (-61.4 %) followed by basalt coarse (-21.4%), produced less $CO_2$ than the control sites, thereby acting as more effective treatment to enhance alkalinity (Fig. 7b).





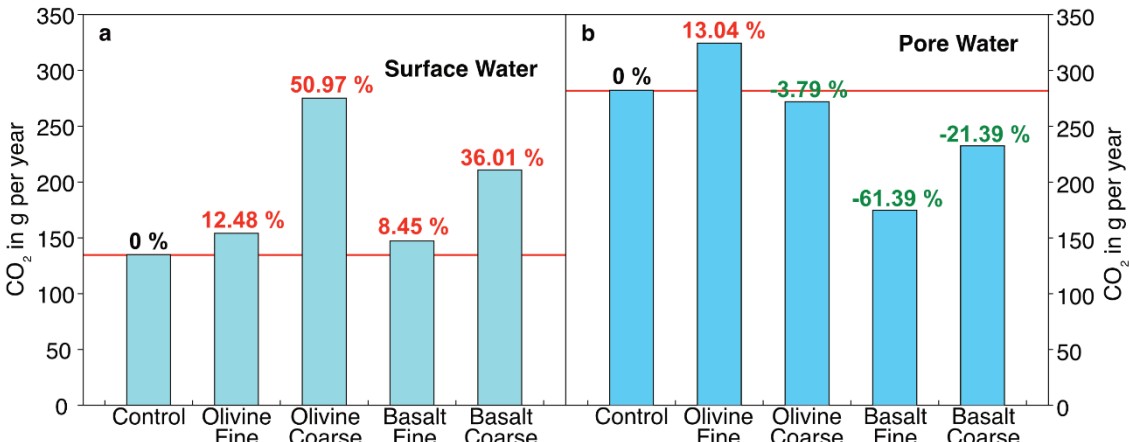

**Figure 7: Carbon dioxide mass (g) in surface (a) and pore water (b) calculated for the first year of the experiment (September 2022 to September 2023), based on the flux measurements in each box.**

## 4 Discussion

### 4.1 Effectiveness of olivine and basalt for seawater alkalinity enhancement

Total alkalinity increased, in both surface and pore water, the day after olivine and basalt deployment in the pioneer vegetation zone of the Ria Formosa salt marsh (Fig. 3). No statistically significant differences ($p > 0.05$) in TA enhancement were observed between olivine and basalt substrates, which showed a similar pattern during the one-year experiment (Sep. 2022 to Sep. 2023). In surface and pore waters, different effects were observed: in surface water, TA increased to a maximum of 22% (maximum average excess of 0.55 mM for BC) and decreased to levels close to the control just one month after the deployment of substrates. In the pore water, the steep increase of 80% (maximum average excess of 2.3 mM for BF) is followed by an exponential decline during the next six months. In both cases, surface and pore water, from April 2023 onwards, the alkalinity in the treatments with olivine and basalt did not differ significantly ($p > 0.05$) from the untreated control sites (Fig. 3, Supplementary Figs. S1, S2). The alkalinity decrease may be due to the formation of secondary minerals, the precipitation of dissolved weathering products (Montserrat et al, 2017; Rigopoulos et al., 2018; Fuhr et al., 2022; Flipkens et al., 2023; Bach, 2024; Geerts et al., 2025), or seasonal variations in temperature and pH (e.g., Wang and Cai, 2004; Santos et al., 2021).

A laboratory dissolution experiment with fine-grained olivine in artificial seawater, showed the highest TA after one day with a subsequent TA decline after 130 days (Fuhr et al., 2022). The temporal dynamics was attributed to the formation of authigenic precipitates, mainly aragonite and sepiolite. Their formation was accompanied with a drawdown of carbonate ion concentrations in the closed experimental system. Supply of carbonate ions is limited in a confined experiment but sufficient in an open system as our experiment at Ria Formosa. Authigenic minerals were not only formed as precipitates but also as coatings. In particular amorphous Mg-Si phases were recognised on the grains by microprobe analyses (Fuhr et al., 2022; Rigopoulos et al., 2018). These coatings may successively impede the dissolution of mafic minerals and could be responsible



for the exponential decrease in TA production during our experiment. Substrate burial by bioturbation, as indicated by crab or worm holes, and the covering by algae or transported sediment may also play a role in the decreased potential of substrates dissolution. In pore water, the excess alkalinity produced by the fine-grained substrates of olivine and basalt were higher than in the coarse-grained treatments during the first two months. This could be considered as effect of adherent dust from industrial milling of the rocks quarried, which was indicated by laboratory experiments too (Fuhr et al., 2022). The raw materials used

in the experiment were not washed to simulate upscale scenario conditions. This also can explain the "almost immediate" decline of alkalinity in the pore water during the first month after deployment (Fig. 3b).

### 4.2 Salt marsh alkalinity enhancement as a measure for atmospheric $CO_2$ removal

The underlying concept of our approach is that coastal wetlands produce TA and DIC, which is exported to the ocean (Reithmaier et al., 2020; Tamborski et al., 2021). TA is mainly controlled by the presence of bicarbonate ($HCO_3^-$) and carbonate

($CO_3^{2-}$). In the sediments, TA can also be produced under anaerobic conditions in the salt marsh soil by the decomposition of organic matter. It includes sulphide and bicarbonate. Once the sulphide passes the near-surface oxic zone, it is oxidized to sulphate and consumes part of the TA. Therefore, TA:DIC ratios are lower in pore water than in surface water (Reithmaier et al., 2023) (Fig. 4) moreover because the concentration of $CO_2$ increases in the sediments (Wang and Cai, 2004) that is not accounted in the TA calculation. The outwelling is fuelled by tidal pumping and facilitated by conduits created by burrows

penetrating in the anoxic zone (Chen et al., 2021). These natural processes are better observed in the control boxes of our experiment. The TA:DIC ratios were on average 1.15 in the surface water, and 0.89 in the pore water (Fig. 4), as well as 1.11 in the channel water at lowering tide, which is, generally, in agreement with data from other salt marshes (Reithmaier et al., 2023). No seasonal variability in this ratio was detected at Ria Formosa.

Except for September and December 2022, the $CO_2$ fluxes from the pore water to the surface water were higher than the fluxes

from the surface water to the atmosphere in the control boxes (Fig. 5). The fluxes at the sediment-water interface showed a seasonal pattern of variability. They were higher in spring and summer, when the mineralisation of sedimentary organic matter was higher at elevated sediment temperatures (Fig. 2a). This pattern corroborates observations from other areas, where salt marsh soils were considered as significant producers of $CO_2$, even though only a few percent are delivered back to the atmosphere (Reithmaier et al., 2020; Nakamura et al., 2024). The $CO_2$ production and seepage at Ria Formosa was proven by

the markedly lower fluxes at the sediment - water interface in the experimental boxes on the day after mineral deployment, while the flux in the control box was same as high as in the following months (Fig. 5b), suggesting that diffusion and seepage were apparently limited by the layer of substrate applied. The flux of $CO_2$ from the control surface water to the atmosphere, was in September 2022 of 10.06 mmol m$^{-2}$ d$^{-1}$ on average, which is in good agreement with values from a *Spartina alterniflora* vegetated salt marsh at Dafeng Milu National Nature Reserve, Jiangsu Province, China (4.8 mmol m$^{-2}$ d$^{-1}$) (Yau et al., 2022).

The question remains why the $CO_2$ flux from pore water to surface water was consistently higher in olivine (both OF and OC treatments) than in basalt treatments (Fig. 5b, 7). To our best knowledge, the only conceivable explanation is fertilisation or



pH-driven changes in microbial communities (Yu et al., 2022), and a higher metabolic activity in the soil below the substrate cover (Corbett et al., 2024).

During the first day after substrate deployment, the pH of pore water in OF was one unit higher than in the control box
(Supplementary Fig. S4). According with the Bjerrum plot a pH change from 7 to 8 effected a conversion of 19 % of the $CO_2$ to $HCO_3^-$, which raised the TA in the pore water by 2.1 mM as compared to the control (Fig. 3b, Supplementary Fig. S2). The DIC raised too by approximately the same amount (Supplementary Fig. S5). With the information available, it is rather unlikely that the $CO_2$ comes from the soil below the substrate, because flux from pore to surface water for the treatments is close to zero (Fig. 5b). Biological activity and the natural variations exerted a greater influence thereafter. Worm holes and crab
burrows were observed in the substrates one week and three months, respectively, after deployment indicating that $CO_2$ seeping from the anoxic black soil below may also contribute to $HCO_3^-$ formation after one week, or that aerification of the sediments during lower water may have facilitated the diffusion of some $CO_2$ from the sediments directly to the atmosphere. An influence of $CO_2$-rich ebb waters derived from the Faro-northwest waste water treating plant is considered negligible, even though the discharge is at maximum in fall (Sierra et al., 2024). Part of DIC and TA followed the same trend, as the excess TA production
through time and was outwelled. It is important to note that TA and DIC outwelling (Fig. 6) were orders of magnitude higher than either the pore water - surface water or the surface water - atmosphere fluxes (Fig. 5) meaning that despite the saltmarsh area acts as a natural sink for atmospheric $CO_2$, it is the main source to export $CO_2$, regardless of its fate, which remains unclear, specifically when and where the $CO_2$ is transported or removed. It can be driven to channels and ultimately to the adjacent Atlantic Ocean. The DIC outwelling in the first 7 months of the experiment was 3.5 % (93.3 mmol m$^{-2}$ d$^{-1}$) higher in
the treatments than in the control, whereas the TA outwelling was 2.6 % higher than in the control. DIC outwelling of the untreated control boxes was on average 2605 mmol m$^{-2}$ d$^{-1}$ during winter (Nov. 2022 - Feb. 2023) and 2453 mmol m$^{-2}$ d$^{-1}$ in summer (Jun. - Sep. 2023) (Fig. 6b). These fluxes were approximately 4 to 14 times higher than those found in the salt marshes at Dafeng Milu National Nature Reserve, Yancheng, Jiangsu Province, China (Chen et al., 2022: 177-675 mmol m$^{-2}$ d$^{-1}$; Yau et al., 2022: 202 mmol m$^{-2}$ d$^{-1}$), or in a salt marsh at Sage Lot Pond, Waquoit Bay, Massachusetts, United States, which was
also dominated by *Spartina alterniflora* (Wang et al., 2016: 95 mmol m$^{-2}$ d$^{-1}$). These are also higher than those estimated for 14 saltmarshes worldwide (Reithmaier et al., 2023), which ranged from -2 to 1200 mmol m$^{-2}$ d$^{-1}$. In the present study however, the outwelling DIC values were derived from individual boxes (0.36 m$^2$) in the pioneer zone of the salt marsh, colonized by *Spartina maritima*, indicating that this area exports higher fluxes to the ocean as compared with other saltmarsh levels (e.g., Reithmaier et al., 2023).

No significant differences were observed between olivine and basalt treatments in the Ria Formosa alkalinity enhancement in-situ experiment or between the different grain sizes used. This inhibition may be considered as on-site negative emission, which could be scaled and accounted. Nevertheless, in terms of $CO_2$ removal estimates, basalt and particularly the fine grain size (~60% of removal; Fig. 7) appears to be more promising than olivine as a potential substrate for removing atmospheric carbon dioxide in this ecosystem by increasing alkalinity. In the BF treatment, for instance, 360 g $CO_2$ (Fig. 7) were not emitted
due to the deployment of 5500 g substrate. The ratio by weight is 1:15. In terms of financial efforts, the current price is 1.7 €

per 5.5 kg Durubas deployed, and the price for $CO_2$ emission certificates is 0.0198 € per 360 g, hence the ratio is 1:86. Costs for transport, deployment and surveillance and their carbon footprint are not considered in this estimate because they are difficult to constrain. Even with the present figures it is evident however, that seawater alkalinity enhancement for atmospheric $CO_2$ removal in the intertidal zone by enhanced weathering of mafic minerals and rocks is far from profitable unless further

technological progress is made.

**Data availability**

The data that support the findings of this study will be published on Figshare.

**Supplement material**

Supplementary information: Figures S1 to S5 and table S1.

**Author contribution**

I.M., J.L., and J.S. conceptualized the study; I.M. funding acquisition and project administration; I.M. and J.L., participated in all sample collection, with the collaboration of J.S. and A.C.; J.L. built the dataset and performed the calculations with A.C. collaboration; I.M., J.L. and J.S. wrote the original draft with A.C. significant contributions. All the authors revised the manuscript and gave final approval for publication.

**Competing interests**

The authors declare that they have no conflict of interest.

**Acknowledgements**

The authors acknowledge to the Instituto de Conservação da Natureza e das Florestas, DRCNF-Algarve (S-026426/2022) and Agência Portuguesa do Ambiente for authorization to carry out the RECAP experiment. The authors also acknowledge to

Anke Dettner-Schönfeld and team members of the RECAP project Cátia Correia, A. Rita Carrasco, Ana Gomes, Patricia Grasse and Óscar Ferreira for field work participation, Paulo Pedro for collaboration in alkalinity analyses in the Laboratory of Chemical Analysis (University of Algarve), and Margarida Ramires for all the collaboration in cleaning field work material. A special acknowledgment is extended to all volunteers participating in the RECAP field work campaigns.





**Financial support**

Research conducted in the framework of the RECAP project, funded by the Fundação para a Ciência e a Tecnologia (FCT), Portugal, under grant number PTDC/CTA-CLI/1065/2021 (https://doi.org/10.54499/PTDC/CTA-CLI/1065/2021), which also supported J. Lübbers. I. Mendes was supported by the contracts CEECINST/00052/2021/CP2792/CT0012 ((https://doi.org/10.54499/CEECINST/00052/2021/CP2792/CT0012) and 2023.10993.TENURE.029, both funded by FCT. This study was also supported by FCT national funds under the project UIDB/00350/2020

(https://doi.org/10.54499/UIDB/00350/2020) granted to CIMA BASE and project LA/P/0069/2020 (https://doi.org/10.54499/LA/P/0069/2020) granted to the Associate Laboratory ARNET.

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
