# Peer review of "Evaluating ocean alkalinity enhancement for carbon dioxide removal: evidence from a one-year saltmarsh field experiment"

_EGUsphere, 2025_

## Author Comment (AC2)

**Authors response to: RC2**: 'Comment on egusphere-2025-4555', Anonymous Referee #2, 05 Dec 2025

The authors present observations from a year-long field trial of alkalinity enhancement in a salt marsh and use these to evaluate the feasibility of the method in a similar coastal system. The experimental design appears to have been made with care and attention to detail. The paper concludes that large scale deployment of OAE in a saltmarsh system is unlikely to be viable, partly due to the modest benefits that are extrapolated from this small-scale trial to a larger deployment. These results provide needed in-situ context for rapidly evolving discussions about deploying OAE at scale. There is some incomplete reasoning that could be expanded upon and I have made suggestions for changes to some of the figures and data presentation.

We thank the reviewer for their thorough evaluation and constructive comments on our manuscript. We appreciate the positive feedback to the experimental design and implementation, and the value of our experimental results for ongoing discussions about large-scale OAE measures. We will carefully consider the reviewer's recommendations to elaborate on sections of the manuscript where reasoning was incomplete and to improve the clarity of figures and data presentation, when the manuscript is going to be revised.

Major comments:

Scaling to annual values – this seems to be a bit liberal – consider working with daily, or seasonal rates that may be more appropriate to your experimental results?

We thank the reviewer for this thoughtful comment. We understand the concern regarding the potential liberality of scaling our flux estimates to annual values. The reviewer is correct that these estimates are approximations based on flux measurements collected during approximately one quarter of the day, once per month, and within relatively small experimental plot areas (60 cm × 60 cm). As described in Section 2.7 (Carbon dioxide mass calculations), $CO_2$ fluxes were first calculated in mg day$^{-1}$ for each treatment and sampling event; treatment-specific daily averages were then multiplied by 365 to produce annual average values.

We agree that such an extrapolation introduces uncertainties and that daily or seasonal scaling could, in some contexts, more conservatively reflect the temporal coverage of the dataset. However, we chose to report annual values because they are commonly used in the existing literature and facilitate a direct comparison with previously published OAE and coastal carbon flux studies. To address the reviewer's concern, we will note these limitations in the revised manuscript and will clarify the assumptions underlying the annual extrapolation in the Results and Discussion sections.

Figures – it is difficult to follow the results describing the difference (or similarity) between pore water and surface water (and rising and lowering tide) with the axes limits in the current figures. Consider using a range that better suits the data, even if it means changes within a figure (i.e., subplots with different ranges).

We appreciate the reviewer's observation regarding the difficulty of recognising differences between porewater and surface water data across tidal stages under the current axis limits. Our intention in using a consistent y-scale across these comparisons was to facilitate direct visual assessment of the relative magnitudes among surface water, porewater, and rising versus lowering tidal conditions. We recognize, however, that this approach may reduce resolution for parameters with narrower dynamic ranges.

To address this concern, we are going to highlight that the alkalinity data displayed in Figure 3 are also provided as excess alkalinity in Supplementary Figures S1 (surface water) and S2 (porewater), where treatment–control differences are more clearly resolved because the axes are optimized for each dataset. In addition, all underlying data will be made publicly available on Figshare to support an independent evaluation.

If the editor decides that adopting variable y-scales within multi-panel figures would improve clarity without compromising the comparability, we are fully prepared to change the figures accordingly.

Section 3.4 – the last line of this section, which indicates that positive (outgassing) fluxes were observed throughout the experiment, and that olivine treatments were associated with larger fluxes than basalt and the control, seems like an important result and should perhaps be included in the abstract?

We acknowledge the reviewer suggestion. We will incorporate this important result in the revised abstract.

Section 3.5 – the fact that the outwelling (advection) of both DIC and TA is far larger than the local fluxes (pore water to surface water, surface water to atmosphere) seems important, but slightly buried in the manuscript? Would this suggest that a trial with much larger quantities of minerals would be needed to get the right 'signal to noise' in these advection dominated (or tidally-influenced) systems?

This is a misunderstanding, probably induced by a direct comparison of Figures 5 and 6. While Figure 5 describes the flux of $CO_2$ in terms of outgassing from the plots, Figure 6 scales the advection of TA and DIC, including dissolved and dissociated CO2, which is, of course far higher. We are going to emphasize this in the revised manuscript to avoid further confusion.

Minor Comments –

Abstract – the first line states that 'OAE is a carbon dioxide removal strategy aimed at reducing atmospheric CO2' this goes without saying – i.e., it is the definition of CDR.

This is a misunderstanding. The acronym OAE, i.e. Ocean Alkalinity Enhancement, is not provided here in the Abstract but defined in line 25 in the Introduction. We concede that it would be better to mention it in the Abstract already, which we are going to do in the revised manuscript.

OAE and TA are defined several times and only rarely used. Define early in the text and be consistent with the use of the shortform thereafter.

We appreciate the reviewer's observation regarding the repeated definitions and inconsistent use of abbreviations for Ocean Alkalinity Enhancement (OAE) and Total Alkalinity (TA). We will revise the manuscript to define these terms early in the text and ensure consistent use of the acronyms throughout the document.

Line 200 – put the '(39.5)' after 'salinity' at the beginning of this sentence.

Accepted.

Line 255 – 'TA:DIC ration of lowering tide RESEMBLE the ratios of control surface water samples' – can this be quantified – it is not possible to see this from the figures that are referred to (with the current axis limits).

We appreciate the reviewer's observation regarding the TA:DIC ratio during lowering tide and its resemblance to control surface water samples. We agree that quantifying this relationship would improve clarity. Accordingly, we will include the requested quantification in the revised manuscript. As previously mentioned regarding the limitations of the axes, our intention was to enable a direct visual comparison of the water samples.

Line 257 – remove 'throughout the year'

Accepted.

Line 295 – unclear what 'globally' means here?

In this context, the term *"globally"* was intended to mean *"in general"* rather than referring to a worldwide scale. To avoid ambiguity, we will exchange the terms to clarify the meaning in the revised manuscript.